Subject Areas:
ecology/physiology

Keywords:
bees, survival, pesticide residue, vitellogenin, p-coumaric acid, metabolization

Authors for correspondence:
Lena Barascou
e-mail: lena.barascou@inrae.fr
Cedric Alaux
e-mail: cedric.alaux@inrae.fr

# Pollen nutrition fosters honeybee tolerance to pesticides

Lena Barascou[1], Deborah Sene[1], Alexandre Barraud[2], Denis Michez[2], Victor Lefebvre[2], Piotr Medrzycki[3], Gennaro Di Prisco[3,4], Verena Strobl[5], Orlando Yañez[5], Peter Neumann[5], Yves Le Conte[1] and Cedric Alaux[1]

[1]INRAE, Abeilles et Environnement, Avignon, France
[2] Research Institute for Biosciences, Laboratory of Zoology, University of Mons, Place du Parc 20, 7000 Mons, Belgium
[3]Council for Agricultural Research and Economics—Agriculture and Environment Research Centre, Via di Corticella 133, 40128 Bologna, Italy
[4]Institute for Sustainable Plant Protection, National Research-Council, Piazzale Enrico Fermi 1, 80055 Portici, Naples, Italy
[5]Institute of Bee Health, Vetsuisse Faculty, University of Bern, Bern, Switzerland

LB, 0000-0002-2379-7214; DS, 0000-0001-5134-6288

A reduction in floral resource abundance and diversity is generally observed in agro-ecosystems, along with widespread exposure to pesticides. Therefore, a better understanding on how the availability and quality of pollen diets can modulate honeybee sensitivity to pesticides is required. For that purpose, we evaluated the toxicity of acute exposure and chronic exposures to field realistic and higher concentrations of azoxystrobin (fungicide) and sulfoxaflor (insecticide) in honeybees provided with pollen diets of differing qualities (named S and BQ pollens). We found that pollen intake reduced the toxicity of the acute doses of pesticides. Contrary to azoxystrobin, chronic exposures to sulfoxaflor increased by 1.5- to 12-fold bee mortality, which was reduced by pollen intake. Most importantly, the risk of death upon exposure to a high concentration of sulfoxaflor was significantly lower for the S pollen diet when compared with the BQ pollen diet. This reduced pesticide toxicity was associated with a higher gene expression of vitellogenin, a glycoprotein that promotes bee longevity, a faster sulfoxaflor metabolization and a lower concentration of the phytochemical p-coumaric acid, known to upregulate detoxification enzymes. Thus, our study revealed that pollen quality can influence the ability of bees to metabolize pesticides and withstand their detrimental effects, providing another strong argument for the restoration of suitable foraging habitat.

# 1. Introduction

The availability of nutritive resources has long been acknowledged as a key ecological factor affecting the expression of several life-history traits [1]. Notably, the quantity and balance of macro- and micronutrients, as well as secondary metabolites, in the diet of insects, can determine their longevity and ability to respond to environmental pressures, such as xenobiotics [1]. For instance, variation in the protein : carbohydrate ratio can modulate their sensitivity to toxins [2], and secondary metabolites may increase their tolerance to various pesticides by stimulating the production of detoxification enzymes [3,4]. In this context, the contribution of resource availability and quality to the overall health of honeybees (*Apis mellifera*), a major pollinator of crops and wild plants [5], has received increased attention [6–8]. Indeed, like many organisms, their environment has rapidly changed under the influence of human activity. They are thus exposed to more frequent and diverse sources of stress, including pesticides, along with a reduction of floral resource abundance and diversity due to landscape simplification and habitat loss [9].

Among the different stress factors threatening honeybee colonies, pesticides have attracted most of the attention and debate [10–12]. The toxicity of a large range of pesticides has been documented [13–19]. Research on the modulation of pesticide toxicity by nutritional factors, while still in its infancy, could lead to a better understanding of the impact of pesticides on honeybee populations and the design of more supportive habitats. The amount of nutrients in nectar and pollen can indeed differ between plant species (6.3–85% for sugar concentration in nectars [20], and 2.5–61% and 1–20% for protein and lipid contents in pollens, respectively [21,22]). In addition, both pollen and nectar are nutritional sources of several amino acids, minerals, micronutrients and secondary metabolites [8,23]. As a consequence, the quality of honeybee diets varies greatly over time and according to landscape features [24,25]. Therefore, these variations in nutritional content may provide a basis for nutritional modulation of pesticide toxicity.

Confirmation of this hypothesis was tested for nectar by providing bees with limited access or access to low concentrations of sugar. The survival of bees was synergistically reduced by the combination of poor nutrition and field-realistic exposure to neonicotinoids (−50%) [26]. However, most of data on the nutritional modulation of pesticide toxicity come from studies that have tested pollen diets, probably because pollen is essential to the physiological development of bees [6,27–29]. Wahl & Ulm [30] were the first to report that feeding bees with pollen increased their tolerance to pesticides. They notably found that pollen intake as well as the quality of pollen (protein content) increased the median lethal dose ($LD_{50}$) of several pesticides [30]. Later, Schmehl *et al.* [31] demonstrated that pollen intake reduced bee sensitivity to chronic exposure to chlorpyrifos compared with bees fed without pollen. At the same time, Archer *et al.* [32] showed that bees having access to an artificial protein-rich diet were more able to withstand nicotine exposure than bees provided with a protein-poor diet. More recently, Crone & Grozinger [33] found that artificial and pollen diets characterized by different protein to lipid ratios can influence the survival time of bees chronically exposed to chlorpyrifos. Endpoints other than mortality rate have been used to assess the influence of pollen quality and availability on pesticide sensitivity in honeybees (development of feeding glands [34]), as well as in bumblebees (micro-colony performance, nest founding [35–37]) and *Osmia* (reproduction [38]); however, these studies generally reported a lack of interactions between these two factors.

Regarding the potential mechanisms underlying this nutritional modulation of pesticide sensitivity, pollen intake upregulates the expression of several xenobiotic-metabolizing cytochrome P450 genes [31], as well as the activity of glutathione *S*-transferases [39], which are involved in Phases I and II of the detoxification pathways, respectively [40–42]. More specifically, upon ingestion, several constituents of pollen, like the phytochemicals *p*-coumaric acid and quercetin, can upregulate the expression of cytochrome P450 genes [43,44] and increase the survival rate of bees exposed to pesticides [3,45–47]. Such an effect on the detoxification capacity of bees was further confirmed by measuring pesticide metabolism. Ardalani *et al.* [48] found a reduction in the concentration of imidacloprid in honeybee bodies fed with quercetin, although no effect was observed on the reduction in the concentration of tebuconazole or tau-fluvalinate. Similarly, adding *p*-coumaric acid to a sucrose diet led to faster coumaphos disappearance [43]. Overall, these studies indicate that pollen may influence the ability of bees to metabolize pesticides, which was recently confirmed [49]. Lastly, we cannot exclude an influence of pollen on the ability of bees to withstand the effects of pesticides, given that the impact of pesticides depends not only on the fate of the molecule in the body (uptake, distribution, biotransformation, elimination), but also on its interaction with the biological target and effects at the physiological level. Due to its positive effect on bee longevity and on several molecular pathways and physiological

functions (e.g. energy storage, immunocompetence, nutrient metabolism, protection against oxidative stress) [50–53], pollen consumption might help bees to better tolerate the wide range of physiological impairments caused by pesticides, notably on nutrient metabolism, immunity, cell signalling and developmental processes [31,54–57].

In sum, it was found that pollen nutrition can influence the survival rate of pesticide-exposed bees and the metabolization of pesticides. However, the nutritional modulation of pesticide sensitivity and the underlying mechanisms have been rarely studied together. To further examine the influence of pollen on pesticide sensitivity and the underlying mechanisms, we provided bees with pollen of different qualities (protein, lipid, *p*-coumaric acid contents) or no pollen and then exposed them to either sulfoxaflor, a new neurotoxic insecticide that shares the same mode of action with neonicotinoids [58,59], or azoxystrobin, an inhibitor of mitochondrial respiration in fungi and one of the most frequently detected fungicides in pollen collected by bees (34–87.5% of samples) [60]. We then determined whether the survival of bees exposed to a single dose of pesticide (previously identified as the median lethal dose) or chronically to field realistic and higher concentrations of pesticides is affected by pollen intake and/or the quality of pollens. Finally, we investigated the mechanisms underlying the modulation of pesticide sensitivity by testing whether pollen consumption induces a decrease in the concentration of pesticides in bees and/or help to tolerate the detrimental effect of pesticides on bee vitality. The latter was determined by measuring the gene expression level of vitellogenin, a well-established marker of bee health and longevity [61,62].

# 2. Material and methods

## 2.1. Pollen diet quality

In order to assess the influence of pollen intake and pollen quality on bee sensitivity to pesticides, we used two pollen blends that differed regarding their nutritional properties: one predominantly composed of *Brassicaceae* (36%) and *Quercus robur* (35%) (*BQ* pollen), and the other primarily composed of *Salix* (89%) (*S* pollen) (see electronic supplementary material, table S1 for the pollen species composition). They were purchased fresh from Abeille heureuse® (France). We analysed protein and lipid content and their ratio following protocols detailed in [22]. The *BQ* pollen had higher protein and lipid content (respectively, $28.39 \pm 0.72\%$ and $18.7 \pm 1.6\%$, $n = 9$) than the *S* pollen ($21.49 \pm 1.05\%$ and $14.07 \pm 1.5\%$, $n = 9$). The protein : lipid ratio was similar between pollen mixes (*BQ* pollen: 1.52 and *S* pollen: 1.53). We also determined the concentrations of both phytochemicals, *p*-coumaric acid and quercetin (see electronic supplementary material). The *p*-coumaric acid concentrations reached $244.7 \text{ mg kg}^{-1}$ (1491.6 µM) in the *BQ* pollen and $104.5 \text{ mg kg}^{-1}$ (637 µM) in the *S* pollen. The level of quercetin was under the quantification limit of the analysis method for both pollens, i.e. below $10 \text{ mg kg}^{-1}$. The presence of pesticide residues in one extract of each pollen blend was determined by liquid chromatography–tandem mass spectrometry (LC-MS/MS) with a limit of quantification (LOQ) of $0.01 \text{ mg kg}^{-1}$ and a limit of detection of $0.005 \text{ mg kg}^{-1}$ following the European Standard EN 15662:2018 procedure (see electronic supplementary material, table S2 for the list of analysed pesticides). Only residues of 2,4-dimethylformamide (DMF, degradation products of amitraz) and tau-fluvalinate were detected in both pollen blends but were below the LOQ. These compounds used as chemical treatments against the honeybee parasite *Varroa destructor* are consistently found in pollens (47.4% and 88.3% of trapped pollens for amitraz and tau-fluvalinate, respectively; [63,64]) and are considered as relatively safe for honeybees with an oral $LD_{50}$ of 75 µg/bee for amitraz (contact exposure) and 45 µg/bee for tau-fluvalinate (oral exposure) [65]. Both pollen blends were gamma irradiated to avoid parasite contamination and stored at $-20°C$.

## 2.2. Influence of pollen nutrition on honeybee sensitivity to pesticides

Experiments were performed at the Institut National de la Recherche Agronomique (INRAE) in a semi-urban area (Avignon, France, 43°540 N–4°520 E) with honeybees (*Apis mellifera*) from our local apiary. To obtain 1-day-old bees, brood frames containing late-stage pupae were removed from 8 to 10 colonies (depending on the experiments) and kept overnight in an incubator under controlled conditions (34°C, 50–70% relative humidity (RH)). The next day, newly emerged bees (less than 1 day old) were collected, mixed and placed in cages ($10.5 \times 7.5 \times 11.5$ cm) [66]. To better simulate colony

rearing conditions, cages were equipped with Beeboost® (Ickowicz, France), releasing one queen-equivalent of queen mandibular pheromone per day.

Caged bees were kept in an incubator (30°C and 50–70% RH) and provided with water and Candy (Apifonda® + powdered sugar) ad libitum. Except of the control groups, bees were also provided with one of the fresh pollen blends (BQ or S pollens) via an open tube feeder for 7 days. To prevent a potential nutritive compensation of bees fed with one of the pollens, they were not provided with ad libitum pollen, but with a determined quantity of pollen each day, representing the minimal daily consumption of pollen: 4 mg/bee/day for the first 2 days, 5 mg/bee/day for the next 2 days, 3 mg/bee/day for the 5th day and 2 mg/bee/day for the last 2 days, as described in Di Pasquale et al. [39]. If some bees died during the pollen-feeding period (7 days), pollen amount was adjusted to the number of surviving bees. Both pollen diets were fully consumed every day.

### 2.2.1. Acute single exposure

In the first experiment, we tested whether pollen intake and pollen quality could modify the sensitivity of bees to a single dose of pesticide previously identified as an $LD_{50}$ (M Medrzycki, G Di Prisco, V Strobl, O Yañez, P Neumann 2019, unpublished data). Groups of 20 one-day-old bees were placed in cages, which were randomly assigned to the different experimental groups: control (sucrose solution only), BQ pollen, S pollen, azoxystrobin, sulfoxaflor, BQ pollen/azoxystrobin, S pollen/azoxystrobin, BQ pollen/sulfoxaflor and S pollen/sulfoxaflor ($n = 10$ cages per experimental group).

On day 5, bees were sugar-starved for 2 h and then fed with a solution of 50% (w/v) sucrose and azoxystrobin (4600 µg ml$^{-1}$, 1.14% acetone) or sulfoxaflor (3.67 µg ml$^{-1}$, 0.37% acetone) according to the experimental group. Sugar solutions were provided via a feeding tube with a hole at its extremity. Each treated cage received 200 µl of the solution laced with pesticides. Solutions were provided for 4 h and all of them were consumed within this time period. Assuming that the bees equally consumed the solutions, pesticide treatments resulted in a theoretical exposure to 46 µg/bee of azoxystrobin and 36.7 ng/bee of sulfoxaflor, corresponding to the $LD_{50}$ levels previously determined. Control groups were fed with pesticide-free sugar solution (50% w/v sucrose, 1% acetone). After exposure to pesticides, bees were provided with water and Candy (Apifonda® + powdered sugar) ad libitum. Mortality was recorded 48 h after exposure.

Stock solutions of sulfoxaflor (Techlab, France) and azoxystrobin (Sigma Aldrich, France) in acetone were previously aliquoted and conserved at −20°C. The exact concentrations were confirmed with LC-MS/MS (see below) and resulted in 5746 µg ml$^{-1}$ for azoxystrobin and 3.62 µg ml$^{-1}$ for sulfoxaflor, which corresponds to a real exposure of 57.5 µg/bee and 36.2 ng/bee, respectively.

### 2.2.2. Chronic exposure

In the second experiment, bees were chronically exposed to two concentrations of pesticides: a concentration that was considered to be field realistic and a higher concentration representing a worst-case exposure scenario. Groups of 30 one-day-old bees were placed in cages ($n = 10$ cages per experimental group) and treatment groups were provided with one of the pollen blends as described above. On day 5, caged bees were provided with a solution of 50% (w/v) sucrose, 0.1% acetone and azoxystrobin or sulfoxaflor at either a low or high environmental concentration which corresponded to theoretical values of 0.02 and 2 µg ml$^{-1}$ for sulfoxaflor and 0.2 and 2 µg ml$^{-1}$ for azoxystrobin according to the experimental group. Control groups were fed with pesticide-free sugar solution (50% w/v sucrose, 0.1% acetone). The concentrations were chosen based on pesticide residue data found in pollen and nectar. Different application rates of sulfoxaflor before or during flowering of cotton resulted in 6.6–13.8 ppb of sulfoxaflor in nectar and 7.7–39.2 ppb in pollen of cotton flowers [67]. However, other field residue studies with cotton, buckwheat and phacelia reported higher levels of sulfoxaflor ranging from 0.05 to 1 ppm in nectar and from 0.22 to 2.78 ppm in pollen collected by honeybees during the flowering period [68,69]. Azoxystrobin was found at levels ranging from 0.03 to 0.11 ppm in pollen collected by honeybees in North America [64]. In France, these levels ranged from 0.01 to 1.9 ppm (Observatory of Pesticide Residue, ITSAP—Institut de l'Abeille 2014, personal communication). The chronic pesticide treatments were performed over 10 days and the syrup feeders were replaced every day. For each cage, individual syrup consumption was assessed daily by weighing feeders and dividing the consumed food by the number bees remaining alive. The cumulated syrup consumption over the 10 days of exposure to pesticide was then determined. After exposure to pesticides, bees were provided with water and Candy (Apifonda® + powdered sugar) ad

libitum. Dead bees were counted daily and removed over a 16-day period (i.e. when the high sulfoxaflor concentration group reached 100% mortality). Following the chemical analyses (LC-MS/MS), the real concentrations of tested diets resulted in 0.02 and 2.35 µg ml$^{-1}$ for sulfoxaflor and 0.22 µg ml$^{-1}$ and 1.90 µg ml$^{-1}$ for azoxystrobin, respectively, for the low and high exposure rates.

## 2.3. Potential mechanisms underlying the nutritional modulation of pesticide sensitivity

In order to investigate the mechanisms underlying the beneficial effect of pollen on pesticide sensitivity, we compared the gene expression level of vitellogenin and the amount of pesticide among groups. Groups of 80 one-day-old bees were placed in cages and, as above, fed with one of the pollen diets ($n = 10$ cages per experimental group). On day 5, bees were sugar-starved for 2 h and then fed with the highest concentration of azoxystrobin and sulfoxaflor (2 µg ml$^{-1}$), or sugar solution only. Each cage received 800 µl of sugar solution. Solutions were provided for 4 h and all of them were consumed within this time period, giving a theoretical dose of 20 ng of pesticide per bee (19 ng of azoxystrobin and 23.5 ng of sulfoxaflor based on the real concentration of the tested solution, see above). After exposure to pesticides, bees were provided with water and Candy (Apifonda® + powdered sugar). At 8 and 24 h post-exposure (i.e. once all the solutions were consumed), 25 and 35 bees per cage were, respectively, sampled on dry ice and stored at −80°C for later analysis.

### 2.3.1. Influence of pollen nutrition and pesticides on vitellogenin expression level

For each cage, the abdomens of six bees sampled at 24 h post-exposure were pooled in groups of three. Abdomen pools were homogenized in 800 µl of Qiazol reagent (Qiagen) with a Tissue Lyser (Qiagen) (4 × 30 s at 30 Hz). RNA extraction was then carried out as indicated in the RNeasyPlus Universal kit (Qiagen) with DNase treatment (Qiagen). RNA yields were measured with a Nanodrop (Thermo Scientific) and cDNA synthesis was carried out on 1 µg of RNA per sample using the High capacity RNA to cDNA kit (Applied Biosystems®, Saint Aubin, France). cDNA samples were diluted 10-fold in molecular grade water. The expression level of vitellogenin was determined by quantitative PCR using a Step One-Plus Real-Time PCR System (Applied Biosystems) and the SYBR green detection method. Three microlitres of cDNA were mixed with 5 µl SYBR Green Master Mix, 1 µl of forward primer (10 µmol) and 1 µl of reverse primer (10 µmol) of the target gene. A dissociation stage for the subsequent melting curve analysis was included. All qPCR reactions were run in duplicate. The average cycle threshold values of vitellogenin were normalized to the geometric mean of the housekeeping genes actin and *RPS18*, which proved to have rather stable expression levels [70]. We used sequences of primers previously published [71,72]. The ΔCt value of each group was subtracted by the ΔCt value of the control group (sugar syrup only) to yield ΔΔCt.

### 2.3.2. Influence of pollen nutrition on pesticide detoxification

Pesticide concentrations were analysed on a pool of 25 bees per cage and time point in the following groups: sulfoxaflor, azoxystrobin, *BQ* pollen/sulfoxaflor, *BQ* pollen/azoxystrobin, *S* pollen/sulfoxaflor, *S* pollen/azoxystrobin.

Sulfoxaflor and azoxystrobin content were analysed via LC-MS/MS. The QuEChERS method was used for the extraction of the active ingredients from samples, following the European Standard EN 15662. Briefly, samples were ground in liquid nitrogen and 2 g of the crushed sample was mixed with 15 ml of a 1 : 2 water and acetonitrile mixture and a bag containing 4 g of magnesium sulfate, 1 g of sodium chloride, 1 g of sodium citrate tribasic dihydrate and 0.5 g of sodium citrate dibasic sesquihydrate. An aliquot of the supernatant was mixed with 900 mg of magnesium sulfate, 150 mg of PSA and 150 mg of C18-E. After centrifugation, 2 µl of extract was injected into an Accela 1250 ultra-high performance liquid chromatography (UHPLC) system for sulfoxaflor or azoxystrobin detection. The UHPLC system was coupled to a TSQ Quantum Access MAX Triple-Stage Quadrupole Mass Spectrometer, equipped with a heated-electrospray ionization (H-ESI) source working in positive polarity. The mobile phase used for the analysis consisted of 4 mM ammonium formate in water and 4 mM ammonium formate in MeOH, both containing 0.1% formic acid. The fragments analysed were at $m/z$ 372.1; 329.1; 344.1 (products) generated by the ion at $m/z$ 404.12 (parent, azoxystrobin), and the fragments at $m/z$ 154.1 and 104.2 (products) generated by the ion at $m/z$ 278.1 (parent, sulfoxaflor). Quantification was performed using acetamiprid as an internal standard. The LOQ for azoxystrobin and sulfoxaflor was 0.001 mg kg$^{-1}$ and 0.01 mg kg$^{-1}$, respectively.

## 2.4. Statistical analysis

Data were analysed using the statistical software R v. 3.3.3 [73]. In the acute toxicity tests, the percentage of dead bees in each cage was determined and each cage was considered as a replicate. Since data were not normally distributed, the effect of pesticide and pollen treatments on bee mortality was analysed using Kruskal–Wallis, followed by Dunn's multiple comparison tests with the Benjamini–Hochberg correction. Then, the epsilon squared ($\varepsilon^2$) was computed to obtain a measure of effect size between experimental groups (*epsilonSquared* function of the *rcompanion* package [74]). The interpretation values were as follows: $\varepsilon^2 < 0.01$: very small effect, $0.01 < \varepsilon^2 < 0.08$: small effect, $0.08 < \varepsilon^2 < 0.26$: medium effect and $\varepsilon^2 \geq 0.26$: large effect [75]. Survival data from the chronic toxicity tests were analysed with a Cox proportional hazards regression model (*coxph* function of the *survival* package in R [76]). Data were transformed in a survival table and the remaining bees were considered alive at day 16. The Cox model was used to calculate the hazard ratio (HR). The HR is defined as the ratio between the instantaneous risk in the treatment group ($H_1$) and the risk in the control group ($H_0$), occurring at a given time interval [77]. The influence of experimental treatments on the cumulated syrup consumption, vitellogenin expression level and pesticide detoxification was analysed using Kruskal–Wallis, followed by Dunn's multiple comparison tests with the Benjamini–Hochberg correction.

# 3. Results

## 3.1. Influence of pollen nutrition on honeybee sensitivity to pesticides

### 3.1.1. Acute single exposure

The dose of azoxystrobin significantly increased the mortality of bees that did not have access to pollen (Kruskal–Wallis test: $p < 0.05$, and Dunn *post hoc* tests: $p < 0.05$; figure 1*a*). Although only 10% of bees were found dead 48 h post-exposure (versus 0% in control groups), the size of the negative effect of azoxystrobin could be considered as medium ($\varepsilon^2 = 0.163$; table 1). Azoxystrobin did not increase bee mortality in bees fed with either the *BQ* or *S* pollen diets (Dunn *post hoc* tests: $p = 0.41$ and $p = 0.79$ for *BQ* and *S* pollen, respectively; figure 1*a*). However, bee mortality did not differ between the different pollen diets (no pollen, *BQ* and *S* pollen) after exposure or not to azoxystrobin (figure 1*a*).

Sulfoxaflor increased the mortality of bees over 48 h (Kruskal–Wallis test, $p < 0.01$; figure 1*b*). For instance, the dose of sulfoxaflor killed around 50% of the bees that did not ingest pollen (figure 1*b*). However, sulfoxaflor toxicity was also reduced by pollen consumption. First, the sulfoxaflor-induced mortality was significantly lower in bees fed with *BQ* or *S* pollen diets than in bees fed without pollen (*BQ* pollen: $p < 0.05$ and *S* pollen: $p < 0.05$). Second, the sulfoxaflor toxicity was reduced by half in bees provided with pollen (*BQ* pollen: $\varepsilon^2 = 0.183$, *S* pollen: $\varepsilon^2 = 0.143$—medium effect) when compared with bees fed without pollen ($\varepsilon^2 = 0.277$—large effect; table 1).

Ultimately, the two types of pollen diet did not differentially affect the acute toxicities of azoxystrobin and sulfoxaflor (*BQ* pollen versus *S* pollen: $p = 1.0$ for azoxystrobin and sulfoxaflor; figure 1*a,b*).

### 3.1.2. Chronic exposure

In the azoxystrobin experiment, we found that, regardless of exposure to pesticides, bees provided with pollen consumed more syrup than bees who did not receive pollen (Kruskal–Wallis test: $p < 0.001$; figure 2*a*). However, besides the non-intoxicated bees who consumed less syrup than intoxicated bees in the *BQ* pollen groups, there was no difference in syrup consumption between pesticide treatments for a given pollen diet. In the sulfoxaflor experiment, the pollen effect on syrup consumption was only found in non-intoxicated bees: bees without pollen consumed less syrup than bees with pollen (Kruskal–Wallis test: $p < 0.001$; figure 2*b*). At the low and high concentration of sulfoxaflor, pollen diets did not affect syrup consumption. The main variation in syrup consumption was due to the high concentration of sulfoxaflor. Bees exposed to 2 ppm of sulfoxaflor consumed less syrup than bees exposed to 0 or 0.02 ppm of sulfoxaflor (although it was not significant for the *BQ* pollen groups).

Chronic exposure to azoxystrobin (0.2–2 ppm) did not affect bee survival whether bees consumed pollen or not (Cox model, $p = 0.17$; figure 3*a*). However, both sulfoxaflor concentrations decreased bee survival (Cox model, $p < 0.001$, figure 3*b*). While, in bees fed without pollen, the highest concentration of sulfoxaflor (2 ppm) caused 100% bee mortality within 16 days, the lowest concentration (0.02 ppm) reduced the survival probability by around 25%.

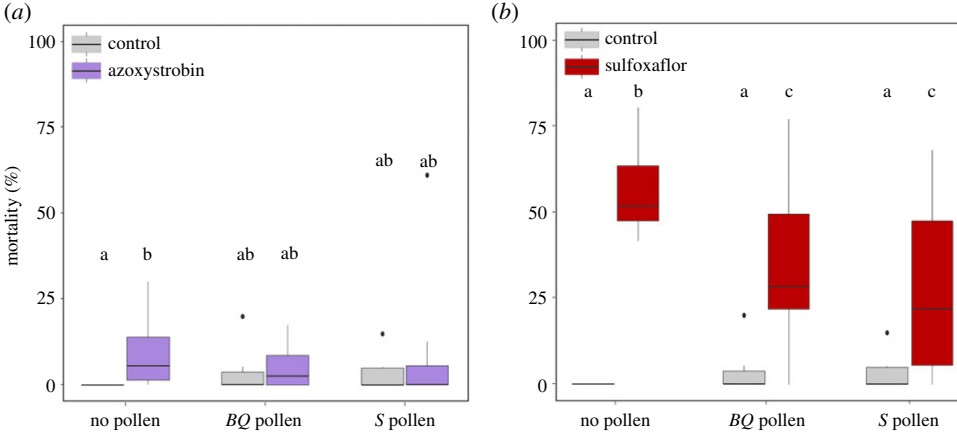

**Figure 1.** Acute toxicity of azoxystrobin (57.5 µg/bee) (*a*) and sulfoxaflor (36.2 ng/bee) (*b*) on bees fed with different pollen regimes. Data represent the 48 h post-exposure mortality of bees (*n* = 20 bees per cage and 10 cages per modality). Boxes indicate the first and third interquartile range with a line denoting the median. Whiskers include 90% of the individuals, beyond which circles represent outliers. Different letters indicate significant differences (Kruskal–Wallis tests followed by Dunn's multiple comparison test).

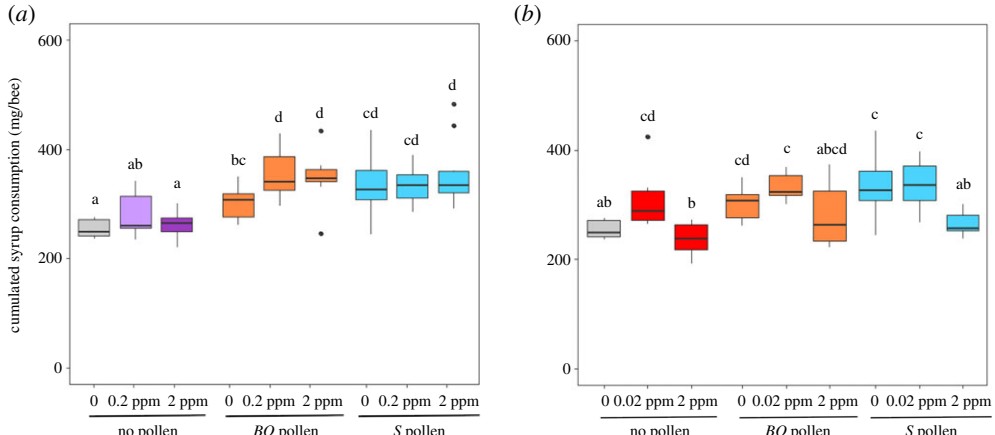

**Figure 2.** Individual syrup consumption according to pesticides and pollen-feeding treatments. Cumulative individual consumption (mg/bee) is shown for bees exposed to azoxystrobin (*a*) and sulfoxaflor (*b*) (*n* = 30 bees per cage and 10 cages per experimental conditions). Boxes indicate the first and third interquartile range with a line denoting the median. Whiskers include 90% of the individuals, beyond which circles represent outliers. Different letters indicate significant differences (Kruskal–Wallis tests followed by Dunn's multiple comparison test).

**Table 1.** Effect size ($\varepsilon^2$) of pesticide acute toxicity. The interpretation values are as follows: $\varepsilon^2 < 0.01$: very small effect, $0.01 < \varepsilon^2 < 0.08$: small effect, $0.08 < \varepsilon^2 < 0.26$: medium effect and $\varepsilon^2 \geq 0.26$: large effect. 95% CI, 95% confidence interval; *p*-values are derived from *post hoc* Dunn tests.

| comparison | $\varepsilon^2$ | 95% CI inf | 95% CI sup | *p*-value |
|---|---|---|---|---|
| control versus azoxystrobin | 0.163 | 0.059 | 0.301 | <0.05 |
| *BQ* pollen versus *BQ* pollen + azoxystrobin | 0.011 | $4.01 \times 10^{-5}$ | 0.086 | 0.12 |
| *S* pollen versus *S* pollen + azoxystrobin | 0.001 | $3.73 \times 10^{-5}$ | 0.117 | 0.70 |
| control versus sulfoxaflor | 0.277 | 0.149 | 0.395 | <0.001 |
| *BQ* pollen versus *BQ* pollen + sulfoxaflor | 0.183 | 0.065 | 0.313 | <0.001 |
| *S* pollen versus *S* pollen + sulfoxaflor | 0.143 | 0.031 | 0.273 | <0.001 |

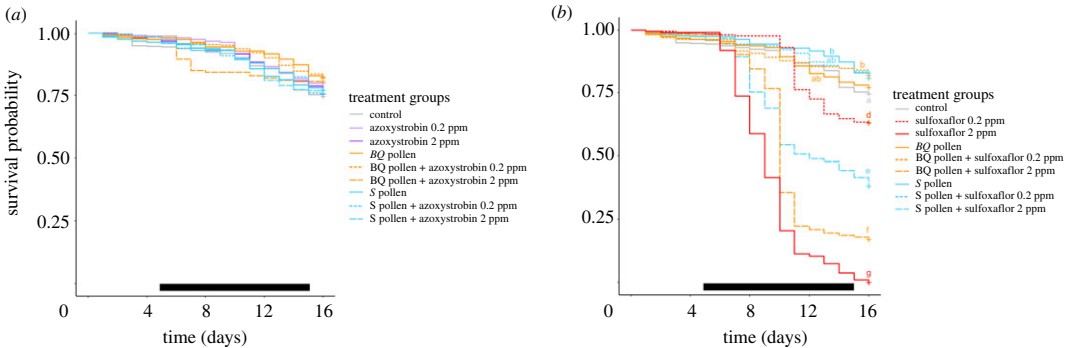

**Figure 3.** Chronic toxicity of environmentally relevant concentrations of azoxystrobin (*a*) and sulfoxaflor (*b*) on bees fed with different pollen regimes. Data represent the survival probabilities of bees (*n* = 30 bees per cage and 10 cages per modality). Different letters indicate significant differences (Cox model) and the black bar represents the period of exposure to pesticides.

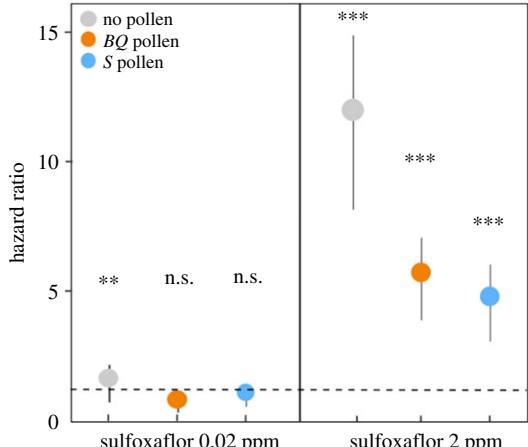

**Figure 4.** Hazard ratio for bees exposed to sulfoxaflor under different pollen-feeding regimes. Bars indicate the 95% confidence interval. Asterisks indicate statistically significant risks of death caused by the pesticide (*$p < 0.05$; **$p < 0.01$ and ***$p < 0.001$) and the dotted line represents HR = 1.

The survival probability of bees over 16 days was enhanced by the *S* pollen (no pollen, $p < 0.05$), but not by the *BQ* pollen ($p = 0.58$), although no difference in survival was found between the two pollen diets ($p = 0.14$; figure 3*b*). The survival probability of bees intoxicated with the low concentration of sulfoxaflor improved with pollen feeding (sulfoxaflor versus sulfoxaflor + *BQ* pollen: $p < 0.001$, sulfoxaflor versus sulfoxaflor + *S* pollen: $p < 0.001$), but no difference was found between the two pollen diets (sulfoxaflor + *S* pollen versus sulfoxaflor + *BQ* pollen: $p = 0.68$). As a consequence, if sulfoxaflor (0.02 ppm) increased the risk of death in bees fed without pollen (HR = 1.53), feeding bees *BQ* pollen or *S* pollen reversed this risk (*BQ* pollen: HR = 0.76; *S* pollen: HR = 1.07; figure 4).

Similarly, the consumption of pollen lowered the negative effect of the highest concentration of sulfoxaflor (sulfoxaflor versus sulfoxaflor + *BQ* pollen: $p < 0.001$ and sulfoxaflor versus sulfoxaflor + *S* pollen: $p < 0.001$). However, the improvement of bee survival was significantly higher when bees consumed the *S* pollen when compared with the *BQ* pollen (sulfoxaflor + *BQ* pollen versus sulfoxaflor + *S* pollen: $p < 0.001$; figure 3*b*). Overall, the consumption of *BQ* pollen and *S* pollen decreased the mortality risk by 2- and 2.5-fold, respectively (*BQ* pollen: HR = 5.72, *S* pollen: HR = 4.79) compared with bees fed without pollen (HR = 12.01; figure 4).

## 3.2. Potential mechanisms underlying the nutritional modulation of pesticide sensitivity

### 3.2.1. Influence of pollen nutrition and pesticides on vitellogenin expression level

The expression level of vitellogenin was significantly affected by the different treatments (Kruskal–Wallis test, $p < 0.001$; figure 5*a,b*). In bees not exposed to pesticide, pollen feeding increased vitellogenin

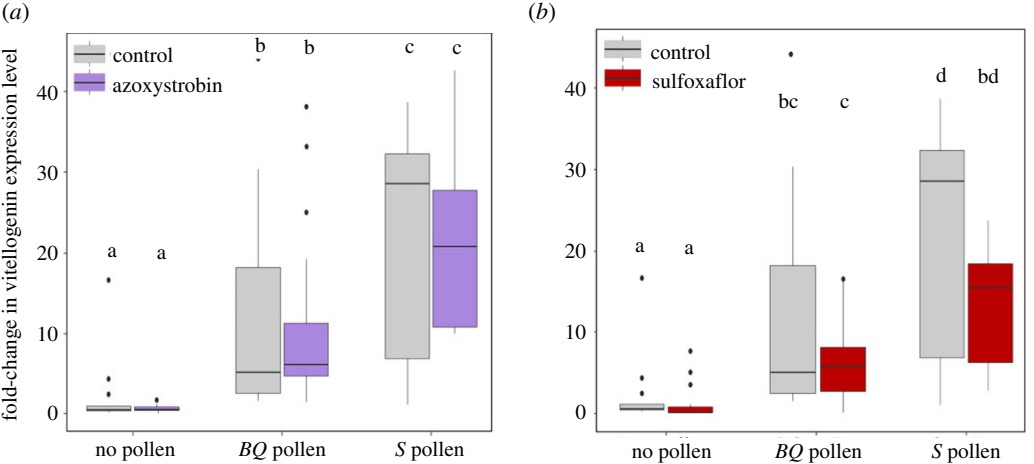

**Figure 5.** Gene expression levels of vitellogenin in response to pesticides and pollen-feeding regimes. Vitellogenin expression levels are shown for bees exposed to azoxystrobin (2 ppm) (*a*) and sulfoxaflor (2 ppm) (*b*) and according to the pollen diets (*n* = 18–20 pools of three bees per experimental condition). Boxes indicate the first and third interquartile range with a line denoting the median. Whiskers include 90% of the individuals, beyond which circles represent outliers. Different letters indicate significant differences (Kruskal–Wallis tests followed by Dunn's multiple comparison tests).

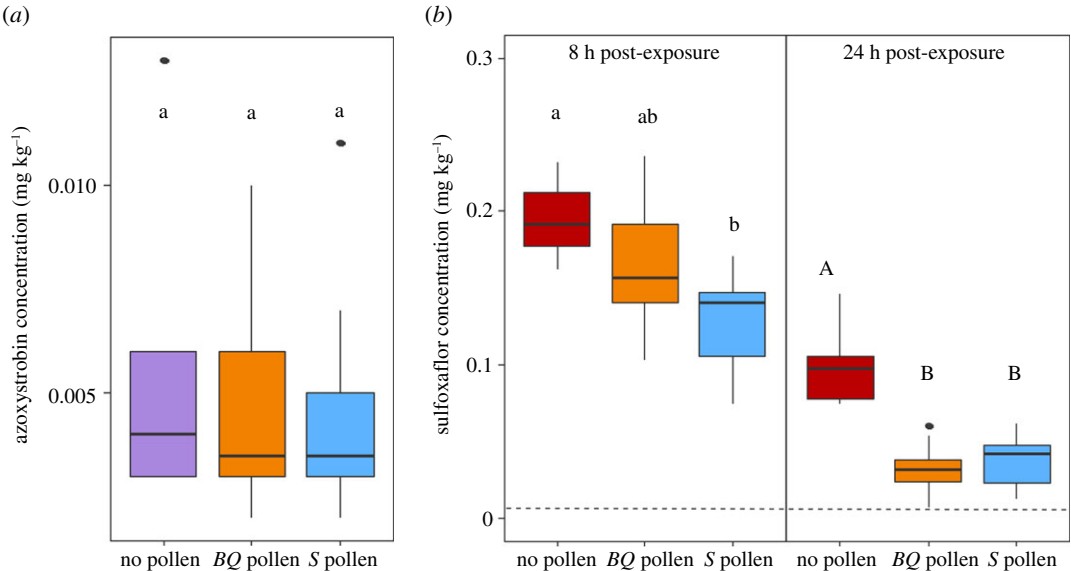

**Figure 6.** Concentration of (*a*) azoxystrobin and (*b*) sulfoxaflor in bees fed with different pollen regimes. Data represent the pesticide concentrations in 10 pools of 25 bees per experimental conditions. Boxes indicate the first and third interquartile range with a line denoting the median. Whiskers include 90% of the individuals, beyond which circles represent outliers. Different letters indicate significant differences (Kruskal–Wallis tests followed by Dunn's multiple comparison tests) and the dotted line represents the LOQ.

expression but the effect was significantly stronger with the *S* pollen (approx. 30-fold) than with the *BQ* pollen (approx. 8-fold, $p < 0.001$).

In all pollen treatments, we did not find any effect of azoxystrobin and sulfoxaflor exposure on vitellogenin expression levels (no pollen, *BQ* pollen and *S* pollen: $p = 1.0$). However, after both pesticide exposures, the level of vitellogenin remained significantly higher in bees fed with the *S* pollen when compared with bees who consumed the *BQ* pollen ($p < 0.05$).

### 3.2.2. Influence of pollen nutrition on pesticide detoxification

Residues of azoxystrobin were detected at very low concentrations at 8 h post-exposure in all treatment groups (figure 6*a*). No significant difference was observed between experimental groups (Kruskal–Wallis test, $p = 0.71$). Since azoxystrobin concentrations were close to the LOQ at 8 h post-exposure, samples at 24 h post-exposure were not analysed.

Regarding sulfoxaflor, the concentrations of residues found in bees 8 h post-exposure were significantly different between experimental groups (Kruskal–Wallis test, $p < 0.01$; figure 6b). The maximum concentrations were found in bees fed only with sugar syrup ($0.19 \pm 0.02$ mg kg$^{-1}$). Consumption of S pollen significantly decreased sulfoxaflor concentrations ($0.13 \pm 0.03$ mg kg$^{-1}$) compared with bees who did not ingest pollen (1.5 times less, $p < 0.01$). The concentrations of sulfoxaflor in BQ pollen-fed bees were intermediate ($0.17 \pm 0.04$ mg kg$^{-1}$) and did not differ from control ($p = 0.11$) and S pollen-fed bees ($p = 0.11$). For each treatment group, the concentration of sulfoxaflor significantly decreased between 8 and 24 h post-exposure (Mann–Whitney test, $p < 0.001$ for each treatment group). It also differed between treatment groups at 24 h post-exposure (Kruskal–Wallis test, $p < 0.001$). Sulfoxaflor concentration was more than two times lower in bees fed with the S or BQ pollen diets than in bees fed without pollen (S pollen: $0.04 \pm 0.02$ mg kg$^{-1}$, BQ pollen: $0.03 \pm 0.02$ mg kg$^{-1}$, and control: $0.10 \pm 0.03$ mg kg$^{-1}$; $p < 0.001$ for both diets). Finally, no difference in sulfoxaflor concentration was found between the two pollen diets at 24 h post-exposure ($p = 0.63$).

## 4. Discussion

In the present study, we showed that pollen consumption, besides its well-known positive effect on honeybee longevity [39,78], can reduce the mortality risk caused by pesticides across different conditions of exposure. In addition, we found that the quality of pollen diets can substantially affect the toxicity of pesticides. These results may help to explain the variability of responses often observed at a given dose or concentration of pesticide [79].

Similar to Wahl & Ulm [30], the negative effect of an acute dose of pesticide was reduced by pollen consumption. The tested dose of azoxystrobin was found to be non-lethal (over 48 h) to bees fed with pollen, while it slightly increased the mortality level of bees fed without pollen. Interestingly, the LD$_{50}$ of azoxystrobin determined during preliminary assays appeared to be less toxic here, providing another example of response variability to pesticides. Experiments were performed in different years and with different colonies (bee genetics), which probably explains the different responses across LD$_{50}$ experiments [80]. Contrary to Wahl & Ulm [30], pollen quality did not influence the sensitivity of bees to the tested doses of pesticides. This lack of effect here might be due to the doses or the pollen diets we used. For instance, measurements should be repeated over a range of dosages to derive more general conclusions about a potential influence of pollen quality. It is also possible that the differences in our pollen diets were not strong enough to influence bee sensitivity to pesticides in the short term (i.e. over 48 h). Similar differences in the nutritional quality of pollens were previously found to affect the chronic susceptibility of honeybees to a parasitic infection [39], suggesting that effects might rather be observed over the long term as confirmed by our chronic exposure experiment.

In the chronic exposure experiment, pollen diets increased the consumption of sugar syrup. Such results are consistent with previous studies, which showed that in response to pollen nutrients, genes involved in carbohydrate metabolism are upregulated [50,81]. This may reflect a higher energy demand since pollen consumption stimulates tissue growth (e.g. hypopharyngeal glands and fat body) [6], which is an energetically costly process. However, this phenomenon was not observed in bees exposed to sulfoxaflor. Syrup consumption did not differ between pollen groups and thus bees provided with different pollen diets were exposed to similar amounts of pesticides. Only bees exposed to the high concentration of sulfoxaflor (2 ppm) tended to consume less syrup. There is now strong evidence for preference or avoidance of sugar syrup laced with pesticides, and this food choice was found to be dependent on pesticide concentration [82,83]. Even though bees are capable of taste perception [84], the underlying mechanisms of food choice are not clearly known [82]. In our experiments, bees were not provided with a food choice, but it is possible that sulfoxaflor at high concentration gives syrup a bitter taste as previously found with high concentration of nicotine [85], which target nicotinic receptors like sulfoxaflor.

Survival data from the chronic exposure experiment further confirmed the beneficial effect of pollen on tolerance to pesticides: the risk of death upon exposure to the low and high sulfoxaflor concentrations disappeared or was significantly reduced by pollen feeding, respectively. This is in accordance with a previous study, which showed that bees fed over several days with a pollen-based diet exhibit reduced sensitivity to a daily exposure to chlorpyrifos [31]. Both pollen diets contained traces of DMF and tau-fluvalinate (below the LOQ), introducing possible interactive effects with the experimental pesticides [18]. However, this was not the case for azoxystrobin since no toxic effect was found on bee survival. Regarding sulfoxaflor, it may have increased its toxicity but to a small extent, since survival upon exposure to sulfoxaflor remained lower in bees fed without pollen than in bees provided with pollen.

Interestingly, bees fed with the S pollen were less sensitive to the high sulfoxaflor concentration when compared with the BQ pollen diet. This suggests that the quality of pollen diets can also affect their capacity to tolerate chronic exposure to pesticides. The higher protective effect of S pollen might result from an improved physiological state. For instance, regardless of exposure to pesticides, the expression level of vitellogenin was much higher in bees fed with the S pollen when compared with bees provided with the BQ pollen. As a glycoprotein with antioxidant functions that protects bees from oxidative stress [86,87], vitellogenin promotes bee longevity but may also have reduced the effects of sulfoxaflor. Indeed, a recent study found that sulfoxaflor increases the level of reactive oxygen/nitrogen species and thus significantly elevates oxidative stress in bees [88]. The higher vitellogenin expression level induced by the S pollen might have thus contributed to better protect bees from exposure to sulfoxaflor, assuming that changes in transcript levels translated into different protein levels. This latter point is supported by the significant decrease in haemolymph vitellogenin concentration upon inhibition of vitellogenin gene activity via RNA interference [89], and the concomitant change in the gene and protein expression of vitellogenin between nurses and foragers [90,91].

We did not find any pesticide-induced differences in vitellogenin levels between bees, both in the presence or absence of a pollen diet. This suggests that the negative impact of pesticides on bee survival does not involve a decline in vitellogenin level, although we cannot eliminate long-term exposure effects. Reported effects of pesticides on vitellogenin in the literature have been contradictory across studies, ranging from no effects (acute exposure to fipronil and deltamethrin [92]), to increasing (chronic exposure to neonicotinoids [93]) and also inhibitory effects (chronic exposure to imidacloprid [94]). This indicates that effects on vitellogenin may be quite variable and possibly depend upon multiple factors, e.g. age of the bees, season, pesticide type and mode of exposure (acute, chronic).

In response to exposure to dietary toxins, organisms have developed elimination mechanisms (e.g. detoxification) that prevent their accumulation in organs and tissues. How the body is able to handle pesticides can, therefore, affect its pesticide sensitivity. The analysis of pesticide residues showed that azoxystrobin was eliminated much faster than sulfoxaflor (approx. 100-fold difference between the two pesticide concentrations at 8 h post-exposure), even though the same doses were given to bees. The mechanisms underlying this faster metabolization of azoxystrobin are not known, but enzymes from the detoxification pathways, like the cytochrome P450 monooxygenases, often exhibit substrate-specificity. For instance, cytochrome P450 members of the CYPQ9 family were found to be responsible for tau-fluvalinate metabolism [44]. Honeybees might thus possess cytochrome P450s that can dock and metabolize azoxystrobin better than sulfoxaflor. This rapid azoxystrobin metabolization might also contribute to its lower toxicity when compared with sulfoxaflor. However, since azoxystrobin is a fungicide, we obviously cannot exclude that it is less efficient in reaching its biological target and/or has a different mode of action in insects.

Finally, sulfoxaflor concentration decreased more quickly in bees fed with the S pollen when compared with bees provided with the BQ pollen. This faster metabolization may participate in the reduced sulfoxaflor toxicity upon ingestion of the S pollen diet. Such results also confirm a recent study, which reported that some pollens are better than others in promoting pesticide metabolization [49]. Different pollens have different nutritional values, which generally translate into differences in bee physiology and longevity [6,39,78,95,96]. Among the pollen nutrients that have positive effects on bee health, the amount of protein plays a substantial role [6,97], although it does not result systematically in healthier bees, especially regarding pathogen resistance [39]. Our study further indicates that the quality of pollen should not only be estimated based on protein content since the S pollen had a lower concentration of protein than the BQ pollen. For instance, the amount of other nutrients, such as amino acids, sterols, vitamins, minerals and nutrient ratio can also influence bee physiology and longevity [98–103]. More specifically, the macronutrients ratio in pollen may also influence the sensitivity of bees to pesticides. This was demonstrated by testing diets with modified protein to lipid ratios (P : Ls) and several pollen diets with different P : Ls [33]. The pollen-induced differences in pesticide sensitivity reported in our study could not be explained by this nutritional factor since both pollen diets had similar P : Ls. However, several studies have shown that the pollen phytochemicals quercetin and p-coumaric acid, upon ingestion, can significantly enhance bee longevity [46,104] and also tolerance to several pesticides [45–47]. However, the effects of these phytochemicals are concentration-dependent; lower concentrations tend to have a positive effect on honeybee longevity (p-coumaric acid at 5, 50 and 500 µM and quercetin at 12.5, 25 and 250 µM), while higher natural concentrations (1000 µM) have no effects [45,46]. Similarly, the reduced mortality risk upon exposure to pesticide was observed over a range of relatively low natural concentrations (5, 50 and 500 µM) for p-coumaric acid. Higher concentrations (1000 µM) increased or did not change the toxicity of pesticides [45]. Our results are, therefore, consistent with these

data given that the $S$ pollen, containing 637 µM of $p$-coumaric acid, was better in improving bee longevity and tolerance to sulfoxaflor when compared with the $BQ$ pollen (1491.6 µM of $p$-coumaric acid). This former concentration might be more optimal for stimulating detoxification enzymes [43] and thereby more quickly eliminating sulfoxaflor, as indicated by our results. It was not possible to quantify quercetin, but its concentration below 33.09 µM probably falls in the range of beneficial concentrations for both pollens, and, therefore, does not explain the differences in pollen quality.

In conclusion, our study demonstrated the modulation of pesticide toxicity by the nutritional state of worker honeybees. Pollen availability and quality, by modifying the physiological background of bees, can improve their ability to eliminate pesticides and withstand their detrimental effects (e.g. protective effect of vitellogenin against oxidative stress), as observed with the high concentration of sulfoxaflor. This nutritional modulation may cause a large range of pesticide responses in the field, given that the abundance and composition of honeybee pollen diets can be highly variable across landscapes and seasons [25,105–110]. A decline in resource availability and biodiversity in agro-ecosystems [111] might, therefore, impair the bee's ability to deal with pesticides [112], giving another strong argument for the restoration of floral resource abundance and diversity in such habitats (introduction of extensive grasslands and flower strips, protection of semi-natural habitats) [113,114]. Further research is, therefore, needed to evaluate the influence of a larger range of pollens of different qualities on pesticide toxicity to better mitigate the impact of exposure to pesticides.

Data accessibility. The data are provided in electronic supplementary material [115].

Authors' contributions. L.B., Y.L.C. and C.A. conceived the study, L.B., D.S., A.B., V.L., G.D.P., V.S., O.Y. and P.M. conducted the experiments, L.B. and C.A. analysed the data, D.M., P.M., P.N., Y.L.C. and C.A. contributed to reagents, L.B. and C.A. wrote the manuscript. All authors read and reviewed the manuscript.

Competing interests. We declare we have no competing interests.

Funding. This project received funding from the European Horizon 2020 research and innovation programme under grant agreement no. 773921 (L.B., Y.L.C. and C.A.).

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
