## [Peer Review File · Royal Society Open Science]

Review History

RSOS-210818.R0 (Original submission)

Review form: Reviewer 1 (Ian Kaplan)

Is the manuscript scientifically sound in its present form?

Yes

Are the interpretations and conclusions justified by the results?

Yes

Is the language acceptable?

Yes

Do you have any ethical concerns with this paper?

No

Have you any concerns about statistical analyses in this paper?

No

Recommendation?

Accept with minor revision (please list in comments)

Comments to the Author(s)

This is a nice study that I enjoyed reading about. I agree with the authors that the interactions between nutrient intake and pesticide tolerance is incredibly important for managing honeybees and other pollinators in agricultural landscapes. This is an area where we have some data emerging, but it is certainly understudied and a topic where we need more data points to make management recommendations.

I liked the authors' approach of using natural pollen mixtures, as opposed to artificially spiking diets with various nutrients. It does limit a bit the interpretation since the ecological difference between a willow vs. oak/Brassica is unclear, in terms of whether that nutritional difference is relevant. Also, I don't think these species are likely to be dominant pollen-types in agricultural areas where the pesticides are being used so this also limits the application of the approach.

The outcomes from the study are interesting and clear. The writing is in fairly good shape throughout. I like the combination of the mechanistic data with the survival data, although quantifying cytochrome p450s would probably be a more direct test of the mechanisms the authors are getting at.

I'm a little unclear on what the outcomes mean for managing floral resources and insecticides in human landscapes. The data from the study indicate that some pollen of any type is necessary to lower pesticide susceptibility. But it would be very unusual if bees had zero access to any pollen, so does that mean that honeybees are always buffered against pesticides in the field (i.e., the no-pollen situation is unrealistic and somewhat irrelevant)? It's possible that pollen buffering is a quantitative phenomenon where periods or landscapes with lots of access to forage provides more buffering than with low access. However, it's hard to make that leap since the authors used a presence/absence design for pollen vs. varying quantity. I think these are considerations worth exploring more in the Discussion.

You may consider checking out several of these papers, some of which are quite recent, that aren't cited in the paper. Not all on honeybees, but certainly still relevant.

Huang Z. 2012. Pollen nutrition affects honey bee stress resistance. *Terrestr. Arthrop. Rev.* 5, 175–189. (doi:10.1163/187498312X639568)

Stuligross C, Williams NM. 2020. Pesticide and resource stressors additively impair wild bee reproduction. *Proc. R. Soc. B* 287, 20201390. (doi:10.1098/rspb.2020.1390)

Klaus F, Tschardt T, Bischoff G, Grass I. 2021. Floral resource diversification promotes solitary bee reproduction and may offset insecticide effects – evidence from a semi-field experiment. *Ecol. Lett.* 24, 668–675. (doi:10.1111/ele.13683)

There are several mistakes in the supplemental paragraph on chemical methods. There are no line numbers so I can't note the location, but please give that another close read.

L24 (and 109). Not sure what you mean by "environmental and higher concentrations"

L28. Increased by how much? Please provide some measure of magnitude.

L30. This is a confusing way to word it. Which diet resulted in lower risk of death upon insecticide exposure? This just says one of the two but doesn't tell you which.

L78. Many will not understand why you're referencing "hypopharyngeal gland development" here or how that relates.

L79. "But overall, studies reported a lack of interactions between these two factors". I'm confused by this sentence. In the paragraph leading up to this sentence you highlight examples where diet

interacts with pesticide exposure, but now you're saying that they don't interact? I think there should be a more consistent message here on what's known.

L129-137. This issue of ambient pesticide residues is an important one. I would need more information here to evaluate. Which specific pesticides were tested? How many samples were analyzed?

L140-145. It would help here if you provided some information on the bees' prior exposure to pesticides and nutrients during development. I realize this is difficult to say but even knowing where these hives were located (i.e., the landscape) provides some context for how much they were pre-exposed to pesticides and which flower types, since this will undoubtedly affect your results.

L150 and 167. How exactly were the pollen and sugar solution administered? Some specifics on the feeding stations would help.

L168. Why "eventually"?

L171. I think this is a big assumption - that all 20 bees shared the solution equally - but there's no real way to control for this variation.

L255. What was considered a replicate in your analyses? I assume the cage was used as a replicate and the 20 bees within the cage were combined for a single value? Clarification here would help. Also, on the statistical front - you have several factors that are fully crossed in a factorial design (pesticide type, pesticide concentration, pollen type). In these circumstances, it's usually more powerful to test the main and interactive effects of each factor, rather than doing a one-way ANOVA-style analysis where all treatments are treated equally. I realize you're using non-parametric analyses, which limits your options, but there are other approaches using GLMs that would account for this. I don't think this will necessarily change your conclusions or outcomes, but there are definitely areas where it would've helped me as a reader. For example, Fig 2 is hard to interpret with all these pairwise comparisons and non-linearities. In cases like this, being able to tell me which of the 3 factors were significant and which factors interacted with the others would make it easier.

L366-367. This is confusing because several lines earlier (L357-358) you stated that the quality of pollen affected toxicity. However, perhaps earlier you're referring to this as a general statement from other studies? I would avoid this since the opening sentence of the Discussion is stating your specific findings so this can get confusing.

L370. Delete "possibly"

L369-371. I agree with this statement. It's hard for the reader to know what these pollen differences represent in terms of quality differences. If anything can be pulled from the literature to support whether this is a small or big difference in protein, etc. in the context of normal honeybee diets, that would help interpretation a lot.

L386. from not form

Figure 3. This is hard to see. At the bare minimum I would make this much larger. It's a lot of treatments to throw on one survival curve panel.

Review form: Reviewer 2

Is the manuscript scientifically sound in its present form?

Yes

Are the interpretations and conclusions justified by the results?

Yes

Is the language acceptable?

Yes

Do you have any ethical concerns with this paper?

No

Have you any concerns about statistical analyses in this paper?

No

Recommendation?

Accept with minor revision (please list in comments)

Comments to the Author(s)

The paper entitled "Pollen nutrition fosters honeybee tolerance to pesticides" by Barascou et al. reports a very interesting study that investigates whether the availability and quality of pollen diets can modulate honeybee sensitivity to the fungicide azoxystrobin and the insecticide sulfoxaflor. Overall, the questions asked in this study are interesting, important and very topical, the methods are adequate and the results are clearly presented. For these reasons, I only have minor comments and suggestions:

Line 108: Please use "identified" instead of "identify"

Line 129: The list of pesticide residues analysed in the different pollen blends should be reported in the supplementary materials.

Line 136: The LD50 values mentioned here for amitraz and tau-fluvalinate refer to the short-term poisoning potential (acute toxicity) of these compounds. However, in this study, pollen blends were consumed by bees for at least 7 days, so they were chronically exposed to both amitraz and tau-fluvalinate. I think this issue should be mentioned in the discussion, and also the possible interactive effects of amitraz and tau-fluvalinate with the pesticides experimentally applied here (sulfoxaflor and azoxystrobin).

Line 141: Here the authors state that honeybees used in this experiment were *Apis mellifera ligustica* x *Apis mellifera mellifera*, how do they know? Have their local apiaries been genetically characterized? If not, I think it would be better to just mention that the experiment was performed with *Apis mellifera*.

Line 162: If LD50 data is not shown here because it will be part of a future publication, I think it would be better to say (data not published), so the reader knows that the data will be available soon.

Line 168: Please briefly explain why the percentage of acetone is different for the azoxystrobin, sulfoxaflor and control sugar solutions.

Lines 189 and 211: Please mention here if pesticide-free syrup contained acetone. Also, please consider using always the same term for the sugar/sucrose solution/syrup when the substance used is the same, in order to avoid confusion.

Line 264: Please write "considered" instead of "considering"

Line 272: Please write "honeybee" instead of "honey bee" to be consistent with the rest of the manuscript.

Line 386: Please change "form" to "from"

Line 399: It may be worth discussing here which component/s or feature/s of *Salix* pollen not measured in this study could be behind its effect on vitellogenin expression levels, and thus, on an improved physiological state of bees. For instance, *Salix* pollen has been shown to contain a relatively high concentration of essential amino acids, sterols and to be highly digestible (Vanderplanck et al. 2016. *Insect Science*). Interestingly, high concentrations of polypeptides/total amino acids and sterols in pollen have been shown to improve the development and performance of other bee species (bumblebees: Vanderplanck et al., 2014. *PLoSOne*).

Finally, it may also be worth clarifying in the discussion that, in this study, bees were exposed to the tested pesticides just through "nectar", overlooking pesticide ingestion via pollen consumption. Exposure via pollen in combination with exposure via nectar would seem to be more field-realistic, and could reveal different results to the ones reported here.

Decision letter (RSOS-210818.R0)

Dear Miss Barascou

On behalf of the Editors, we are pleased to inform you that your Manuscript RSOS-210818 "Pollen nutrition fosters honeybee tolerance to pesticides" has been accepted for publication in Royal Society Open Science subject to minor revision in accordance with the referees' reports. Please find the referees' comments along with any feedback from the Editors below my signature.

Please submit your revised manuscript and required files (see below) no later than 7 days from today's (ie 05-Jul-2021) date. Note: the ScholarOne system will 'lock' if submission of the revision is attempted 7 or more days after the deadline. If you do not think you will be able to meet this deadline please contact the editorial office immediately.

on behalf of Professor Kevin Padian (Subject Editor)
openscience@royalsociety.org

Reviewer comments to Author:

Reviewer: 1
Comments to the Author(s)

This is a nice study that I enjoyed reading about. I agree with the authors that the interactions between nutrient intake and pesticide tolerance is incredibly important for managing honeybees and other pollinators in agricultural landscapes. This is an area where we have some data emerging, but it is certainly understudied and a topic where we need more data points to make management recommendations.

I liked the authors' approach of using natural pollen mixtures, as opposed to artificially spiking diets with various nutrients. It does limit a bit the interpretation since the ecological difference between a willow vs. oak/Brassica is unclear, in terms of whether that nutritional difference is relevant. Also, I don't think these species are likely to be dominant pollen-types in agricultural areas where the pesticides are being used so this also limits the application of the approach.

The outcomes from the study are interesting and clear. The writing is in fairly good shape throughout. I like the combination of the mechanistic data with the survival data, although quantifying cytochrome p450s would probably be a more direct test of the mechanisms the authors are getting at.

I'm a little unclear on what the outcomes mean for managing floral resources and insecticides in human landscapes. The data from the study indicate that some pollen of any type is necessary to lower pesticide susceptibility. But it would be very unusual if bees had zero access to any pollen, so does that mean that honeybees are always buffered against pesticides in the field (i.e., the no-pollen situation is unrealistic and somewhat irrelevant)? It's possible that pollen buffering is a quantitative phenomenon where periods or landscapes with lots of access to forage provides more buffering than with low access. However, it's hard to make that leap since the authors used a presence/absence design for pollen vs. varying quantity. I think these are considerations worth exploring more in the Discussion.

You may consider checking out several of these papers, some of which are quite recent, that aren't cited in the paper. Not all on honeybees, but certainly still relevant.
 Huang Z. 2012. Pollen nutrition affects honey bee stress resistance. *Terrestr. Arthrop. Rev.* 5, 175-189. (doi:10.1163/187498312X639568)
 Stuligross C, Williams NM. 2020. Pesticide and resource stressors additively impair wild bee reproduction. *Proc. R. Soc. B* 287, 20201390. (doi:10.1098/rspb.2020.1390)
 Klaus F, Tschardt T, Bischoff G, Grass I. 2021. Floral resource diversification promotes solitary bee reproduction and may offset insecticide effects – evidence from a semi-field experiment. *Ecol. Lett.* 24, 668-675. (doi:10.1111/ele.13683)

There are several mistakes in the supplemental paragraph on chemical methods. There are no line numbers so I can't note the location, but please give that another close read.

L24 (and 109). Not sure what you mean by "environmental and higher concentrations"

L28. Increased by how much? Please provide some measure of magnitude.

L30. This is a confusing way to word it. Which diet resulted in lower risk of death upon insecticide exposure? This just says one of the two but doesn't tell you which.

L78. Many will not understand why you're referencing "hypopharyngeal gland development" here or how that relates.

L79. "But overall, studies reported a lack of interactions between these two factors". I'm confused by this sentence. In the paragraph leading up to this sentence you highlight examples where diet interacts with pesticide exposure, but now you're saying that they don't interact? I think there should be a more consistent message here on what's known.

L129-137. This issue of ambient pesticide residues is an important one. I would need more information here to evaluate. Which specific pesticides were tested? How many samples were analyzed?

L140-145. It would help here if you provided some information on the bees' prior exposure to pesticides and nutrients during development. I realize this is difficult to say but even knowing where these hives were located (i.e., the landscape) provides some context for how much they were pre-exposed to pesticides and which flower types, since this will undoubtedly affect your results.

L150 and 167. How exactly were the pollen and sugar solution administered? Some specifics on the feeding stations would help.

L168. Why “eventually”?

L171. I think this is a big assumption - that all 20 bees shared the solution equally - but there’s no real way to control for this variation.

L255. What was considered a replicate in your analyses? I assume the cage was used as a replicate and the 20 bees within the cage were combined for a single value? Clarification here would help. Also, on the statistical front - you have several factors that are fully crossed in a factorial design (pesticide type, pesticide concentration, pollen type). In these circumstances, it’s usually more powerful to test the main and interactive effects of each factor, rather than doing a one-way ANOVA-style analysis where all treatments are treated equally. I realize you’re using non-parametric analyses, which limits your options, but there are other approaches using GLMs that would account for this. I don’t think this will necessarily change your conclusions or outcomes, but there are definitely areas where it would’ve helped me as a reader. For example, Fig 2 is hard to interpret with all these pairwise comparisons and non-linearities. In cases like this, being able to tell me which of the 3 factors were significant and which factors interacted with the others would make it easier.

L366-367. This is confusing because several lines earlier (L357-358) you stated that the quality of pollen affected toxicity. However, perhaps earlier you’re referring to this as a general statement from other studies? I would avoid this since the opening sentence of the Discussion is stating your specific findings so this can get confusing.

L370. Delete “possibly”

L369-371. I agree with this statement. It’s hard for the reader to know what these pollen differences represent in terms of quality differences. If anything can be pulled from the literature to support whether this is a small or big difference in protein, etc. in the context of normal honeybee diets, that would help interpretation a lot.

L386. from not form

Figure 3. This is hard to see. At the bare minimum I would make this much larger. It’s a lot of treatments to throw on one survival curve panel.

Reviewer: 2

Comments to the Author(s)

The paper entitled “Pollen nutrition fosters honeybee tolerance to pesticides” by Barascou et al. reports a very interesting study that investigates whether the availability and quality of pollen diets can modulate honeybee sensitivity to the fungicide azoxystrobin and the insecticide sulfoxaflor. Overall, the questions asked in this study are interesting, important and very topical, the methods are adequate and the results are clearly presented. For these reasons, I only have minor comments and suggestions:

Line 108: Please use “identified” instead of “identify”

Line 129: The list of pesticide residues analysed in the different pollen blends should be reported in the supplementary materials.

Line 136: The LD50 values mentioned here for amitraz and tau-fluvalinate refer to the short-term poisoning potential (acute toxicity) of these compounds. However, in this study, pollen blends were consumed by bees for at least 7 days, so they were chronically exposed to both amitraz and tau-fluvalinate. I think this issue should be mentioned in the discussion, and also the possible interactive effects of amitraz and tau-fluvalinate with the pesticides experimentally applied here (sulfoxaflor and azoxystrobin).

Line 141: Here the authors state that honeybees used in this experiment were *Apis mellifera ligustica* x *Apis mellifera mellifera*, how do they know? Have their local apiaries been genetically characterized? If not, I think it would be better to just mention that the experiment was performed with *Apis mellifera*.

Line 162: If LD50 data is not shown here because it will be part of a future publication, I think it would be better to say (data not published), so the reader knows that the data will be available soon.

Line 168: Please briefly explain why the percentage of acetone is different for the azoxystrobin, sulfoxaflor and control sugar solutions.

Lines 189 and 211: Please mention here if pesticide-free syrup contained acetone. Also, please consider using always the same term for the sugar/sucrose solution/syrup when the substance used is the same, in order to avoid confusion.

Line 264: Please write “considered” instead of “considering”

Line 272: Please write “honeybee” instead of “honey bee” to be consistent with the rest of the manuscript.

Line 386: Please change “form” to “from”

Line 399: It may be worth discussing here which component/s or feature/s of Salix pollen not measured in this study could be behind its effect on vitellogenin expression levels, and thus, on an improved physiological state of bees. For instance, Salix pollen has been shown to contain a relatively high concentration of essential aminoacids, sterols and to be highly digestible (Vanderplanck et al. 2016. *Insect Science*). Interestingly, high concentrations of polypeptides/total amino acids and sterols in pollen have been shown to improve the development and performance of other bee species (bumblebees: Vanderplanck et al., 2014. *PLoSOne*).

Finally, it may also be worth clarifying in the discussion that, in this study, bees were exposed to the tested pesticides just through “nectar”, overlooking pesticide ingestion via pollen consumption. Exposure via pollen in combination with exposure via nectar would seem to be more field-realistic, and could reveal different results to the ones reported here.

===PREPARING YOUR MANUSCRIPT===

If you have been asked to revise the written English in your submission as a condition of publication, you must do so, and you are expected to provide evidence that you have received language editing support. The journal would prefer that you use a professional language editing service and provide a certificate of editing, but a signed letter from a colleague who is a native speaker of English is acceptable. Note the journal has arranged a number of discounts for authors

using professional language editing services
(<https://royalsociety.org/journals/authors/benefits/language-editing/>).

===PREPARING YOUR REVISION IN SCHOLARONE===

-- If you have uploaded ESM files, please ensure you follow the guidance at <https://royalsociety.org/journals/authors/author-guidelines/#supplementary-material> to include a suitable title and informative caption. An example of appropriate titling and captioning may be found at https://figshare.com/articles/Table_S2_from_ls_there_a_trade-

off_between_peak_performance_and_performance_breadth_across_temperatures_for_aerobic_sc
ope_in_teleost_fishes_/3843624.

Author's Response to Decision Letter for (RSOS-210818.R0)

See Appendix A.

Decision letter (RSOS-210818.R1)

Dear Miss Barascou,

I am pleased to inform you that your manuscript entitled "Pollen nutrition fosters honeybee tolerance to pesticides" is now accepted for publication in Royal Society Open Science.

on behalf of Prof Kevin Padian (Subject Editor)
openscience@royalsociety.org

Institut National de Recherche en Agronomie
UR 406 Abeilles et Environnement
Domaine St Paul, Site Agroparc
228 route de l'Aérodrome
84914 Avignon
France

Léna Barascou
lena.barascou@inrae.fr

Avignon, July 9th, 2021

Revision of manuscript RSOS-210818

Dear editor(s) of *Royal Society Open Science*,

We were delighted that our manuscript has been accepted for publication in your journal subject to minor revision according to the referees' comments. This was our pleasure because the reviewers have made constructive and helpful comments, which have improved the quality of our manuscript. Please find an account of revisions.

Please find attached our revised manuscript as well our detailed responses to all point raised in **bold** below.

We hope that our manuscript is now up to standards for publication in *Royal Society Open Science*, we look forward to hearing from you soon.

Best wishes
Léna Barascou, PhD student.

Reviewer #1:

Comments to the Author(s)

This is a nice study that I enjoyed reading about. I agree with the authors that the interactions between nutrient intake and pesticide tolerance is incredibly important for managing honeybees and other pollinators in agricultural landscapes. This is an area where we have some data emerging, but it is certainly understudied and a topic where we need more data points to make management recommendations.

We are delighted about this positive feedback.

I liked the authors' approach of using natural pollen mixtures, as opposed to artificially spiking diets with various nutrients. It does limit a bit the interpretation since the ecological difference between a willow vs. oak/Brassica is unclear, in terms of whether that nutritional difference is relevant. Also, I don't think these species are likely to be dominant pollen-types in agricultural areas where the pesticides are being used so this also limits the application of the approach.

We agree that the ecological difference between our pollen diets could be more relevant. However, the scope of our study was to test the availability and quality of pollen diets on the sensitivity of bees to pesticides, and this is what we have shown with S pollen diet that reduced pesticide toxicity better than BQ pollen.

The outcomes from the study are interesting and clear. The writing is in fairly good shape throughout. I like the combination of the mechanistic data with the survival data, although quantifying cytochrome p450s would probably be a more direct test of the mechanisms the authors are getting at.

We appreciate this kind comment and agree that quantifying cytochrome P450s would have been an interesting genes to test. However, to study pesticide detoxification, we decided to use a more direct approach, which is the quantification of pesticide residue. Indeed, it is not clear which P450s is involved in the elimination of sulfoxaflor and azoxystrobin.

I'm a little unclear on what the outcomes mean for managing floral resources and insecticides in human landscapes. The data from the study indicate that some pollen of any type is necessary to lower pesticide susceptibility. But it would be very unusual if bees had zero access to any pollen, so does that mean that honeybees are always buffered against pesticides in the field (i.e., the no-pollen situation is unrealistic and somewhat irrelevant)? It's possible that pollen buffering is a quantitative phenomenon where periods or landscapes with lots of access to forage provides more buffering than with low access. However, it's hard to make that leap since the authors used a presence/absence design for pollen vs. varying quantity. I think these are considerations worth exploring more in the Discussion.

Yes, pollen availability modulates pesticide sensitivity, but we also showed that pollen quality can differentially affect bee sensitivity to pesticides. This is why we concluded on the importance on maintaining floral biodiversity in landscapes.

The influence of pollen quality is reported in the discussion, but to remind the role of pollen quality, we changed "Pollen nutrition" by "Pollen availability and quality" in the conclusion (line 470) and changed the text to "... giving another strong argument for the restoration of floral resource abundance and diversity in such habitats (introduction of extensive grasslands and flower strips, protection of semi-natural habitats)" in lines 476-477.

You may consider checking out several of these papers, some of which are quite recent, that aren't cited in the paper. Not all on honeybees, but certainly still relevant. Huang Z. 2012. Pollen nutrition affects honey bee stress resistance. *Terrestr. Arthrop. Rev.* 5, 175–189. (doi:10.1163/187498312X639568)

Stuligross C, Williams NM. 2020. Pesticide and resource stressors additively impair wild bee reproduction. *Proc. R. Soc. B* 287, 20201390. (doi:10.1098/rspb.2020.1390)

Klaus F, Tschardt T, Bischoff G, Grass I. 2021. Floral resource diversification promotes solitary bee reproduction and may offset insecticide effects – evidence from a semi-field experiment. *Ecol. Lett.* 24, 668–675. (doi:10.1111/ele.13683)

Huang et al is a review with one paragraph on honeybee sensitivity to pesticides citing a single study: Walm and Ulm 1983. Therefore, we would prefer to directly crediting the original study.

We have added Stuligross et al in line 80 and Klaus et al in line 476.

There are several mistakes in the supplemental paragraph on chemical methods. There are no line numbers so I can't note the location, but please give that another close read.

We apologize for this rather sloppy mistake. We now went through carefully through this paragraph and have corrected all typos accordingly.

L24 (and 109). Not sure what you mean by “environmental and higher concentrations”

We have replaced “environmental” by “field realistic” (lines 24, 112).

L28. Increased by how much? Please provide some measure of magnitude.

We have now added “by 1.5 to 12-fold” (line 29).

L30. This is a confusing way to word it. Which diet resulted in lower risk of death upon insecticide exposure? This just says one of the two but doesn't tell you which.

We agree and have reworded the sentence accordingly:

“Most importantly, the risk of death upon exposure to a high concentration of sulfoxaflor was significantly lower for the *S* pollen diet as compared to the *BQ* pollen diet.” (lines 30-31).

L78. Many will not understand why you're referencing “hypopharyngeal gland development” here or how that relates.

Yes, we agree and have changed it to “development of feeding glands” (line 79).

L79. “But overall, studies reported a lack of interactions between these two factors”. I'm confused by this sentence. In the paragraph leading up to this sentence you highlight examples where diet interacts with pesticide exposure, but now you're saying that they don't interact? I think there should be a more consistent message here on what's known.

The lack of interaction was observed for endpoints others than bee survival. We have now merged and rephrased the two sentences to avoid any confusion: “Endpoints other than mortality rate have been used to assess the influence of pollen quality and availability on

pesticide sensitivity in honeybees (development of glands producing larval food [34]), as well as in bumblebees (micro-colony performance, nest founding [35–37]) and Osmia (reproduction [38]), however, these studies generally reported a lack of interactions between these two factors. ” (lines 77-81)

L129-137. This issue of ambient pesticide residues is an important one. I would need more information here to evaluate. Which specific pesticides were tested? How many samples were analyzed?

We have added the list of analyzed pesticides in the supplementary information (lines 135-136). One pollen extract per diet was analysed, we added it in lines 132-133.

L140-145. It would help here if you provided some information on the bees' prior exposure to pesticides and nutrients during development. I realize this is difficult to say but even knowing where these hives were located (i.e., the landscape) provides some context for how much they were pre-exposed to pesticides and which flower types, since this will undoubtedly affect your results.

This makes perfect sense to us. We have therefore added the GPS coordinates of our apiaries and the type of landscape: semi-urban area (line 145). We are sorry but we do not have the flower composition of the landscape.

L150 and 167. How exactly were the pollen and sugar solution administered? Some specifics on the feeding stations would help.

We agree and have specified these points (line 155; lines 172-173).

L168. Why “eventually”?

This word has been deleted (line 171).

L171. I think this is a big assumption - that all 20 bees shared the solution equally - but there's no real way to control for this variation.

We agree and this is also why we used large pools of bees for the analysis.

L255. What was considered a replicate in your analyses? I assume the cage was used as a replicate and the 20 bees within the cage were combined for a single value? Clarification here would help. Also, on the statistical front – you have several factors that are fully crossed in a factorial design (pesticide type, pesticide concentration, pollen type). In these circumstances, it's usually more powerful to test the main and interactive effects of each factor, rather than doing a one-way ANOVA-style analysis where all treatments are treated equally. I realize you're using non-parametric analyses, which limits your options, but there are other approaches using GLMs that would account for this. I don't think this will necessarily change your conclusions or outcomes, but there are definitely areas where it would've helped me as a reader. For example, Fig 2 is hard to interpret with all these pairwise comparisons and non-linearities. In cases like this, being able to tell me which of the 3 factors were significant and which factors interacted with the others would make it easier.

Yes, the percentage of dead bees in each cage was determined and each cage was considered as a replicate (we added it in lines 262 – 263). We agree that multiple factorial

analysis would have been more appropriate, but data were not normally distributed and hence GLM analysis do not seem appropriate to our number of replicates. Regarding the figure 2, we understand it might be difficult to have an overall view of the results, but in the text we specifically focused on the comparisons of interest (between pollens diets for a given pesticide concentration and between pesticide concentrations for a given pollen diet).

L366-367. This is confusing because several lines earlier (L357-358) you stated that the quality of pollen affected toxicity. However, perhaps earlier you're referring to this as a general statement from other studies? I would avoid this since the opening sentence of the Discussion is stating your specific findings so this can get confusing.

We are referring to our study: we also found that pollen quality influences the sensitivity to pesticides as discussed throughout the discussion. We changed the sentence to “In addition, we found that the quality of pollen diets can substantially affect the toxicity of pesticides.” (line 364)

L370. Delete “possibly”

Done (line 377).

L369-371. I agree with this statement. It's hard for the reader to know what these pollen differences represent in terms of quality differences. If anything can be pulled from the literature to support whether this is a small or big difference in protein, etc. in the context of normal honeybee diets, that would help interpretation a lot.

We have now added a reference showing that such quality differences were found to affect bee susceptibility to a parasitic infection (see lines 378-380).

L386. from not form

Done (line 395).

Figure 3. This is hard to see. At the bare minimum I would make this much larger. It's a lot of treatments to throw on one survival curve panel.

We have now enlarged the figure 2 (we assume you refer to fig. 2) as much as we could. We are aware of the difficulty to read a figure with multiple survival curves, which is why we added the fig. 3 summarizing the risk of death associated to the different pesticides.

Reviewer #2:

Comments to the Author(s)

The paper entitled “Pollen nutrition fosters honeybee tolerance to pesticides” by Barascou et al. reports a very interesting study that investigates whether the availability and quality of pollen diets can modulate honeybee sensitivity to the fungicide azoxystrobin and the insecticide sulfoxaflor. Overall, the questions asked in this study are interesting, important and very topical, the methods are adequate and the results are clearly presented. For these reasons, I only have minor comments and suggestions:

We very much appreciate this very kind and positive feedback of the referee.

Line 108: Please use “identified” instead of “identify”

Done (line 111).

Line 129: The list of pesticide residues analysed in the different pollen blends should be reported in the supplementary materials.

We agree and have now added the list of analyzed pesticides in the supplementary information (lines 136-137).

Line 136: The LD50 values mentioned here for amitraz and tau-fluvalinate refer to the short-term poisoning potential (acute toxicity) of these compounds. However, in this study, pollen blends were consumed by bees for at least 7 days, so they were chronically exposed to both amitraz and tau-fluvalinate. I think this issue should be mentioned in the discussion, and also the possible interactive effects of amitraz and tau-fluvalinate with the pesticides experimentally applied here (sulfoxaflor and azoxystrobin).

We have now discussed this point in lines 399-404.

Line 141: Here the authors state that honeybees used in this experiment were *Apis mellifera ligustica* x *Apis mellifera mellifera*, how do they know? Have their local apiaries been genetically characterized? If not, I think it would be better to just mention that the experiment was performed with *Apis mellifera*.

The referee is correct. We don't have any genetic data and therefore stick to *Apis mellifera*, which is absolutely safe.

Line 162: If LD50 data is not shown here because it will be part of a future publication, I think it would be better to say (data not published), so the reader knows that the data will be available soon.

The LD50 data will be the focus of a future publication written by some co-authors of this paper. Accordingly, we have now reworded “data not shown” into “data not published” (line 165).

Line 168: Please briefly explain why the percentage of acetone is different for the azoxystrobin, sulfoxaflor and control sugar solutions.

The percentage of acetone is different because the initial pesticide concentrations and the following series of dilutions were different between pesticides. We don't think it needs to be added in the text.

Lines 189 and 211: Please mention here if pesticide-free syrup contained acetone. Also, please consider using always the same term for the sugar/sucrose solution/syrup when the substance used is the same, in order to avoid confusion.

We have added details about the pesticide-free syrup (lines 193-194) and replaced “syrup” by “sugar solution” (lines 177, 193, 216).

Line 264: Please write “considered” instead of “considering”

Line 272: Please write “honeybee” instead of “honey bee” to be consistent with the rest of the manuscript.

Line 386: Please change “form” to “from”

All done.

Line 399: It may be worth discussing here which component/s or feature/s of Salix pollen not measured in this study could be behind its effect on vitellogenin expression levels, and thus, on an improved physiological state of bees. For instance, Salix pollen has been shown to contain a relatively high concentration of essential aminoacids, sterols and to be highly digestible (Vanderplanck et al. 2016. Insect Science). Interestingly, high concentrations of polypeptides/total amino acids and sterols in pollen have been shown to improve the development and performance of other bee species (bumblebees: Vanderplanck et al., 2014. PLoSOne).

Thank you for these suggestions. We have now added the possible role of these nutrients and corresponding references in lines 80, 450 and 476.

Finally, it may also be worth clarifying in the discussion that, in this study, bees were exposed to the tested pesticides just through “nectar”, overlooking pesticide ingestion via pollen consumption. Exposure via pollen in combination with exposure via nectar would seem to be more field-realistic, and could reveal different results to the ones reported here.

We definitely agree and this scenario of exposure via both nectar and pollen needs to be investigated. However, we don’t really know the possible outcome of this scenario and therefore prefer to not discuss it here. Note, that we have published a study on the impact of pesticide via pollen consumption (Prado et al 2019, <https://doi.org/10.1016/j.scitotenv.2018.09.102>).